# DNA Methylation Profiles of *PSMA6*, *PSMB5*, *KEAP1*, and *HIF1A* Genes in Patients with Type 1 Diabetes and Diabetic Retinopathy

**DOI:** 10.3390/biomedicines12061354

**Published:** 2024-06-18

**Authors:** Zane Svikle, Natalia Paramonova, Emīls Siliņš, Leonora Pahirko, Līga Zariņa, Kristīne Baumane, Goran Petrovski, Jelizaveta Sokolovska

**Affiliations:** 1Faculty of Medicine, University of Latvia, Jelgavas Street 3, LV 1004 Riga, Latvia; zane.svikle@gmail.com (Z.S.); bigzars@gmail.com (L.Z.); kbaumane75@gmail.com (K.B.); 2Institute of Biology, University of Latvia, Jelgavas Street 1, LV 1004 Riga, Latvia; natalia.paramonova@lu.lv; 3Faculty of Physics, Mathematics and Optometry, University of Latvia, Jelgavas Street 3, LV 1004 Riga, Latvia; emils.silins@lu.lv (E.S.); leonora.pahirko@lu.lv (L.P.); 4Ophthalmology Department, Riga East University Hospital, Hipokrata Street 2, LV 1038 Riga, Latvia; 5Center of Eye Research and Innovative Diagnostics, Department of Ophthalmology, Oslo University Hospital, Institute of Clinical Medicine, Faculty of Medicine, University of Oslo, 0372 Oslo, Norway; goran.petrovski@medisin.uio.no

**Keywords:** diabetic eye disease, epigenetics, proteasome

## Abstract

We explored differences in the DNA methylation statuses of *PSMA6*, *PSMB5*, *HIF1A*, and *KEAP1* gene promoter regions in patients with type 1 diabetes and different diabetic retinopathy (DR) stages. Study subjects included individuals with no DR (NDR, *n* = 41), those with non-proliferative DR (NPDR, *n* = 27), and individuals with proliferative DR or those who underwent laser photocoagulation (PDR/LPC, *n* = 46). DNA methylation was determined by Zymo OneStep qMethyl technique. The methylation of *PSMA6* (NDR 5.9 (3.9–8.7) %, NPDR 4.5 (3.8–5.7) %, PDR/LPC 6.6 (4.7–10.7) %, *p* = 0.003) and *PSMB5* (NDR 2.2 (1.9–3.7) %, NPDR 2.2 (1.9–3.0) %, PDR/LPC 3.2 (2.5–7.1) %, *p* < 0.01) differed across the groups. Consistent correlations were observed between the methylation levels of *HIF1A* and *PSMA6* in all study groups. DNA methylation levels of *PSMA6*, *PSMB5*, and *HIF1A* genes were positively correlated with the duration of diabetes, HbA1c, and albuminuria in certain study groups. Univariate regression models revealed a significant association between the methylation level z-scores of *PSMA6*, *PSMB5*, and *HIF1A* and severe DR (*PSMA6*: OR = 1.96 (1.15; 3.33), *p* = 0.013; *PSMB5*: OR = 1.90 (1.14; 3.16), *p* = 0.013; *HIF1A*: OR = 3.19 (1.26; 8.06), *p* = 0.014). *PSMB5* remained significantly associated with DR in multivariate analysis. Our findings suggest significant associations between the severity of DR and the DNA methylation levels of the genes *PSMA6*, *PSMB5*, and *HIF1A*, but not *KEAP1* gene.

## 1. Introduction

Diabetic retinopathy (DR) is one of the most common complications of diabetes, and it has a significant impact on national and global health. In a US study conducted in 2021, an estimated 9.6 million people were found to have DR, with a prevalence rate of 26.43% among individuals with diabetes [1]. The pathogenetic mechanisms responsible for DR are not yet fully understood. At present, there is a deficiency in identifying susceptibility and probability of early indicators for DR and other microvascular complications of diabetes, and there is also a need for effective treatments to manage the progression of the disease during its initial stages.

Epigenetic alterations influence how genes are expressed without making changes to the underlying DNA sequence, adapting dynamically in response to environmental, developmental, and nutritional signals [2,3]. Recently, research has shown that epigenetic mechanisms play a role in diabetic complications [4], although there is not sufficient data about their implication in DR pathogenesis. The development of DR is multifactorial, and it is known to progress even after euglycemia is achieved [5], suggesting that epigenetics may play a role in the development and progression of DR. Indeed, global DNA methylation status was linked to retinopathy in a case–control study of 168 individuals with type 2 diabetes, demonstrating the strong relationship between DNA methylation and the regulation of gene transcription [6]. Moreover, patients with type 1 diabetes and proliferative DR (PDR) have displayed variations in the patterns of epigenetic factors, and this implies that certain changes in epigenetic markers could potentially serve as predictive indicators for the development of PDR and might be valuable as prospective biomarkers for DR screening [7]. Since disease outcomes are thought to be determined by a combination of genotypes, environmental exposures, and their interactions [8], DNA methylation may represent a potential biomarker of DR [9]. Some of the epigenetic modifications, being reversible, present important prospects for therapeutic intervention [10]. There is intense research interest in the development of non-invasive blood-based biomarkers for DR. However, the limited available data on this topic delay the development of diagnostic and disease-monitoring markers, which are related to the methylation statuses of genes in DR.

The ubiquitin–proteasome system plays a crucial role in regulating cell homeostasis through protein degradation [11]. Dysregulation of this system is implicated in various chronic diseases, including diabetes. Proteasomal gene *PSMA6* has been already identified as a gene with type 1 diabetes susceptibility [12], and *PSMB5* genetic variations were found in association with multiple sclerosis in Latvians [13].

Altered proteasome activity in response to hypoxia and oxidative stress under hyperglycemic conditions is observed in diabetes [14], potentially affecting the degradation of proteins involved in anti-oxidative and anti-hypoxic defense mechanisms, such as the Nrf2/Keap1 and hypoxia-inducible factor-1 (HIF-1) systems [15]. Under normal conditions, Keap1 facilitates the proteasomal degradation of Nrf2, but oxidative stress leads to Keap1 inactivation, stabilizing Nrf2 and promoting its anti-oxidative response [16]. In DR, epigenetic changes in the *KEAP1* gene can result in decreased Nrf2 expression, compromising the anti-oxidative response [17]. HIF-1, a key transcription factor in hypoxia responses, plays a critical role in maintaining oxygen homeostasis [18,19]. While HIF-1 alpha is established as a central stimulator of angiogenesis in the proliferative phase of DR, it plays a protective role during the early stages of DR, exhibiting anti-inflammatory, antiapoptotic, and anti-oxidative effects [20]. Notably, the *HIF-1A* Pro582Ser polymorphism, involved in resistance to hyperglycemia, is associated with protection against severe DR [21]. As the disease progresses, factors such as chronic hyperglycemia leading to the formation of advanced glycation end products (AGEs) and the chronic inflammatory state associated with diabetes can compromise HIF-1 function [22,23]. HIF-1 alpha undergoes hydroxylation and proteasomal degradation under normoxic conditions. However, under hypoxia or prolyl hydroxylase domain inhibition, HIF-1 alpha stabilizes, translocates into the nucleus, and activates downstream genes through hypoxia-responsive elements [24]. In experimental studies, inhibitors of proteasome activity have shown a beneficial impact on the progression of diabetic complications [25,26,27,28]. Noteworthily, there is evidence of distinct promoter methylation patterns in UPS genes in cancer [29,30], and in diabetology, the offspring of mothers with diabetes exhibit the altered methylation of UPS genes [31]. Numerous associations between proteasome system genes and autoimmune diseases, including type 1 diabetes, have previously been published in the Latvian population. Polymorphisms in *PSMA3*, *PSMA6*, and *PSMC6* were found to be associated with type 1 diabetes in Latvian patients [12]. Furthermore, correlations have been established between these proteasome gene polymorphisms and type 1 diabetes-susceptible genes involved in various pathways, including innate and adaptive immunity, antiviral response, insulin signaling, and glucose/energy metabolism, highlighting the complex interplay of these genes in type 1 diabetes pathogenesis [12]. This complex chain of associations highlights the importance of the proteasome system in the development of diabetes and its complications.

These findings underscore the significance of investigating UPS regulation and epigenetic modifications in understanding the DR pathogenesis. However, there is currently no published data on the association between the methylation status of genes associated with the ubiquitin–proteasome system and the hypoxia signaling axis and the severity of DR. Our study aimed to study the differences in the methylation statuses of the promoter parts of *PSMA6*, *PSMB5*, *HIF1A*, and *KEAP1* genes between patients with different DR stages and type 1 diabetes in Latvia.

## 2. Materials and Methods

### 2.1. Patients and Ethics

For the implementation of this study, patients in Latvia with type 1 diabetes were recruited in frames of the LatDiane study (a part of the international InterDiane consortium). Inclusion criteria for the study were as follows: patients with age ≥ 18 years and history and established treatment of type 1 diabetes (defined as an age of diagnosis younger than 40 years, with insulin treatment initiated within one year of diagnosis, and C-peptide levels below 0.3 nmol/L) [32]. Exclusion criteria were treatment with oral hypoglycemic medications for more than one year after diagnosis. The study protocol was approved by the Latvian Central Ethics Committee (clearance Nr. 1/19-10-01, issued on 1 October 2019). The recruitment of the study participants, biobanking, and sample storage were performed in agreement with the procedures of the Genome Database of the Latvian population [33] and are described in more detail in [34]. The study corresponds to the ethical standards defined in the 1964 Declaration of Helsinki and its later amendments. Written informed consent was obtained from all study participants before inclusion in the study.

### 2.2. Clinical Definitions

DR grading was based on the fundus oculi examination carried out by an ophthalmologist. Patients were stratified according to the DR status as follows: no retinopathy (NDR), non-proliferative retinopathy (NPDR), proliferative retinopathy (PDR), and patients after panretinal-laser photocoagulation (LPC). Furthermore, to increase the power of the study, subjects were stratified into three groups: patients with NDR; patients with NPDR; patients with PDR, and those after LPC (PDR/LPC).

The “smokers” group applied to patients currently smoking at least one cigarette per day, and smoking was self-reported in the questionnaire.

Body mass index (BMI) was calculated as weight (kg)/height (m)^2^.

The definition of arterial hypertension was based on blood pressure values (systolic blood pressure ≥ 140 mmHg (18.7 kPa) or diastolic blood pressure ≥ 90 mmHg (12.0 kPa)) and the history of antihypertensive drug usage.

We defined cardiovascular disease as a history of stroke, amputation, peripheral vascular disease, acute myocardial infarction, or coronary bypass/percutaneous transluminal coronary angioplasty.

Diabetic nephropathy was defined as follows: macroalbuminuria or eGFR below 60 mL/min/1.73 m^2^ or treatment with dialysis or kidney transplant. Albuminuria was determined using two out of three urine albumin-to-creatinine ratio measurements in the morning spot urine. The estimated glomerular filtration rate (eGFR) was calculated according to Chronic Kidney Disease Epidemiology Collaboration (CKD-EPI).

### 2.3. Biochemical Parameters

Total cholesterol, high- and low-density lipoprotein cholesterol, triglycerides, glycated hemoglobin (HbA1c), and the albumin/creatinine ratio in urine were measured in certified clinical laboratories.

### 2.4. Sampling of Blood for DNA Extraction and Serum Preparation

For serum preparation, peripheral venous blood was collected. The blood samples were incubated undisturbed for 30 min at room temperature and then centrifuged. The serum was removed from the pellet and transferred into fresh 2 mL tubes, frozen, and stored at −80 °C until analysis. Blood for DNA extraction was collected in EDTA tubes. DNA isolation from whole blood samples using the phenol–chloroform extraction method was carried out in the biobank setting, as previously described [33].

### 2.5. Targeted DNA Methylation Assessment

Bisulfite-free, restriction enzyme-dependent determination of the DNA methylation status in the promoter regions of *PSMA6*, *PSMB5*, *HIF1A*, and *KEAP1* genes was performed using real-time PCR procedure, following the instructions given in the manual (OneStep qMethyl™ Kit (Zymo research, Irvine, CA, USA)). The procedure involves two reactions for each investigated DNA sample: a “Test Reaction” and a “Reference Reaction”. In the Test Reaction, DNA is digested with Methylation Sensitive Restriction Enzymes (MSREs), while DNA in the Reference Reaction remains undigested. Real-time PCR with the SYTO^®^ 9 fluorescent dye (Thermo Fisher Scientific, Waltham, MA, USA) is then used to quantify the difference in the methylation status in both reactions. DNA methylation extent was calculated using threshold cycle or CT values and the following equation: 100 × 2^−ΔCT^, indicating methylation status in percentage values. UCSC Genome Browser on Human ((GRCh38/hg38, https://genome.ucsc.edu/) accessed on 27 October 2023) was used to determine the localization of CpG island and promoter regions of genes of interest. Primer design was performed with the online software Primer3plus (https://www.bioinformatics.nl/cgi-bin/primer3plus/, accessed on 27 October 2023 (Netherlands Bioinformatics Centre, Wageningen, The Netherlands)). To perform quantitative real-time polymerase chain reaction (RT-PCR) with methylation-sensitive restriction enzymes (MSREs), we designed primers that included two or more MSRE targeting sites in the amplicon flanking targeted CpG sites regions related to the promoter parts of the investigated genes (Appendix A). For each sample, 5 μL DNA (4 ng/μL) is added to achieve a final reaction volume of 20 μL (1 ng/μL concentration). Duplicate human methylated and non-methylated DNA standards with control MGMT primers, as well as positive controls, were used in each experiment to verify the accuracy of the reaction performed according to the standard procedure.

### 2.6. Statistical Analysis

Categorical data are presented as frequencies and percentages. Most of the variables analyzed violated the normality assumption (determined by the Shapiro–Wilk test), and continuous data are represented as medians with the interquartile range (Q1–Q3). Biomarker levels between DR groups (NDR, NPDR, PDR/LPC) were compared using the Kruskal–Wallis test. Post hoc analysis was performed using Dunn’s test.

Analysis of covariance (ANCOVA) on ranks was conducted to assess differences in biomarker levels among retinopathy groups adjusted for age, sex, and BMI.

Logistic regression models were employed to evaluate the odds of severe DR (PDR/LPC) compared to patients in NDR and NPDR groups for predictors *PSMA6*, *PSMB5*, *KEAP1*, and *HIF1A* while adjusting for age, sex, smoking, and arterial hypertension.

Correlation analysis between biomarkers among DR groups was conducted using the Spearman correlation coefficient.

A *p*-value of less than 0.05 was considered statistically significant.

All statistical data analysis was performed using Statistical Software R version 4.3.0. ((http://www.r-project.org), accessed on 23 April 2024).

## 3. Results

### 3.1. Characteristics of Cohort

A total of 114 patients were analyzed in the study, 46.50% of them were male. The median age was 39 (30–49) years, duration of diabetes 22 (16–30) years, body mass index (BMI) 24.55 (22.22–27.98) kg/m^2^, and waist/hip ratio 0.86 (0.78–0.93). Among all patients, 72 (63.20%) had arterial hypertension, 22 (19.30%) had diabetic nephropathy, 78 (68.40%) had polyneuropathy, and 11 (9.60%) had cardiovascular diseases. In the entire cohort, the median HbA1c level was 8.75 (8.00–10.10) %, high-density lipoprotein cholesterol measured 1.58 (1.24–1.83) mmol/L, low-density lipoprotein cholesterol measured 2.91 (2.07–3.37) mmol/L, triglycerides measured 1.20 (0.80–1.65) mmol/L, the median estimated glomerular filtration rate (eGFR) was reported at 110.57 (91.15–120.53) mL/min/1.73 m^2^, and the median albuminuria level was 1.10 (0.28–8.33) mg/mmol. The use of angiotensin-converting enzyme inhibitors/angiotensin II receptor blockers (ACEI/ARB) was reported by 43 (37.70%) of the patients, while 21 (18.40%) underwent statin therapy.

### 3.2. Characteristics of Patients with Different Severity of DR

Characteristics of the patients within the groups of retinopathies are demonstrated in Table 1. Patients within PDR/LPC group compared to NDR and NPDR groups had significantly longer diabetes duration, a higher prevalence of arterial hypertension and cardiovascular diseases, elevated serum triglyceride levels, lower eGFR, and a higher frequency of ACEI/ARB usage and statin therapy.

Among patients in NPDR, significantly younger age, higher prevalence of smokers, and higher levels of HbA1c were observed compared to both other groups. Patients in the NDR group had significantly higher BMI and lower levels of albuminuria compared to NPDR and PDR/LPC groups. The prevalence of diabetic nephropathy was statistically significantly higher in the PDR/LPC group compared to the NDR group.

### 3.3. Association of DNA Methylation with DR Severity Stages

The level of methylation (%) in the promoter regions of the genes *HIF1A*, *PSMA6*, *PSMB5*, and *KEAP1* was compared in three study groups NDR, NPDR, and PDR/LPC. As demonstrated in Table 2 and Figure 1, only the levels of the methylation of *PSMA6* (*p* = 0.003) and *PSMB5* (*p* < 0.01) genes differed across the study groups.

### 3.4. Correlations between DNA Methylation Levels and Clinical Parameters

Spearman correlation coefficients were calculated to assess the relationships between the methylation levels of genes *PSMA6*, *PSMB5*, *HIF1A*, and *KEAP1* among themselves and with other clinical parameters within the study groups stratified by DR severity stages. The results for parameters exhibiting significant correlations are summarized in Table 3.

In patients within the NDR group, positive correlations were observed between methylation levels of *HIF1A* and *PSMA6* (R = 0.55, *p* < 0.001), *PSMB5* and *KEAP1* (R = 0.46, *p* = 0.003), and *PSMA6* and *PSMB5* (R = 0.35, *p* = 0.024). Additionally, the duration of diabetes was positively correlated with *HIF1A* gene (R = 0.52, *p* < 0.001) and *PSMA6* gene methylation level (R = 0.43, *p* = 0.005). HbA1c was found to be positively correlated with *PSMB5* gene methylation level (R = 0.44, *p* = 0.005).

In the NPDR group, a positive correlation was observed between methylation levels of genes *HIF1A* and *PSMA6* (R = 0.39, *p* = 0.045).

In the PDR/LPC group, positive correlations were observed between methylation levels of genes *HIF1A* and *PSMA6* (R = 0.45, *p* = 0.002) and *PSMA6* and *PSMB5* (R = 0.40, *p* = 0.006). Moreover, albuminuria was observed to be positively correlated with *PSMA6* (R = 0.45, *p* = 0.008) and *PSMB5* methylation levels (R = 0.45, *p* = 0.008).

Overall, consistent correlation trends were observed exclusively between the methylation levels of genes *HIF1A* and *PSMA6* across all study groups. However, other correlations varied among the study groups.

### 3.5. Association of DNA Methylation with the Presence of DR Using Logistic Regression

Logistic regression models were constructed to examine the association between DNA methylation levels and the presence of severe DR (NDR and NPDR vs. PDR/LPC). All continuous predictors were standardized before analysis. The results are summarized in Table 4. Univariate regression analysis demonstrated significant effects of *PSMA6* (OR = 1.96 (1.15; 3.33), *p* = 0.013), *PSMB5* (OR = 1.90 (1.14; 3.16), *p* = 0.013), and *HIF1A* (OR 3.19 (1.26; 8.06), *p* = 0.014).

Multivariate regression models were fitted separately for each gene methylation level adjusted for age and sex (third column in Table 4) and additionally for smoking and arterial hypertension (fourth column in Table 4). Only *PSMB5* showed a trend towards a statistically significant effect (OR 1.75 (0.98; 3.12), *p* = 0.057) when adjusted for age and sex, and a statistically significant effect (OR 1.57 (1.04; 2.36), *p* = 0.030) when additionally adjusted for smoking and arterial hypertension. *HIF1A* demonstrated a trend towards statistical significance (OR 2.13 (0.95; 4.77), *p* = 0.065) when adjusted for age, sex, smoking and arterial hypertension.

## 4. Discussion

In the present study, DNA methylation profiles of *PSMA6*, *PSMB5*, *KEAP1*, and *HIF1A* genes were analyzed in a Latvian cohort of 114 patients to examine the effect of methylation on different stages of retinopathy and to identify correlations with clinical and biochemical parameters.

According to the comparative analysis of the three study groups (NDR, NPDR, and PDR/LPC), methylation levels of genes *PSMA6* and *PSMB5* were found to be significantly higher in patients with PDR/LPC. In addition, in all studied DR groups, positive correlations were observed between the methylation levels of the *HIF1A* and *PSMA6* genes. According to data reported by Maghbooli et al., an increasing trend in global DNA methylation levels has also been observed with progressing retinopathy stages. In addition, gene promoter methylation levels found by us align with the reported global DNA methylation levels, ranging from 1.79% to 7.35% [6]. Subsequent studies have also consistently identified differential DNA methylation in specific genes associated with DR [35,36,37].

The molecular mechanism underlying increased proteasome gene promoter methylation in patients with severe DR possibly involves the epigenetic regulation of *PSMA6* and *PSMB5*, key subunits of the proteasome complex responsible for protein degradation [11,38]. Several studies have shown that severe oxidative stress induced by hyperglycemia decreases intracellular proteolysis, probably by generating damaged proteins that cannot be easily degraded and by damaging proteasomes [39]. The changes in DNA methylation also can be induced by hyperglycemia [40] and other metabolic by-products of diabetes that contribute to the development of diabetic complications in peripheral organs [41,42]. In the context of PDR, this methylation-induced regulation might impair the normal degradation of cellular proteins, leading to the accumulation of damaged or misfolded proteins [43]. Therefore, we can assume that the observed trend of increasing methylation profiles with advancing retinopathy stages in our study implies a cumulative effect of prolonged hyperglycemia and oxidative stress on the epigenetic regulation of these proteasomal genes. Notably, our cohort study in Latvia revealed a positive correlation between *PSMB5* methylation and HbA1c levels in the NDR group, suggesting a complex interaction involving glycemic control, epigenetic changes, and proteasome function. Methylation alterations under hyperglycemia conditions [40] can impact *PSMB5* transcription and, consequently, impact proteasome function, influencing cellular protein homeostasis [44,45,46]. Interestingly, no correlation between HbA1c and *PSMB5* gene methylation was observed in NPDR and PDR/LPC groups, which could be explained by confounding factors: higher levels of HbA1c, higher prevalence of other complications of diabetes, and more frequent usage of antihypertensive and hypolipidemic treatment in these groups [41,42,43,44,45,46]. Importantly, *PSMB5* gene methylation was identified in the current study as a significant predictor of severe DR and has the potential to become a clinically significant biomarker.

The consistent positive trends in the correlation between *HIF1A* and *PSMA6* methylation levels identified across all study groups in the current study can indicate a common regulatory mechanism. The ubiquitin–proteasome system is crucial for regulating the cellular response to hypoxia by controlling the stability of HIF-1 alpha. In normal oxygen conditions, HIF-1 alpha undergoes hydroxylation, ubiquitination, and degradation [47]; under hypoxic conditions, and reduced hydroxylation and impaired ubiquitin–proteasome degradation result in the stabilization of HIF-1 alpha [48,49]. This stabilized form translocates into the nucleus, thereby transactivating the transcription of downstream genes [50]. Other post-translational modifications, including methylation, can also regulate HIF signaling (reviewed in [51]).

Our results are consistent with previously published data on the utility of various DNA methylation changes in the diagnosis of autoimmune diseases associated with disease activity, progression, and clinical outcome. The promoter hypermethylation and associated silencing of the *C9orf72* gene occur in about 30% of amyotrophic lateral sclerosis and frontotemporal degeneration patients with a favorable prognosis [52]. Methylation levels of the *BDNF* gene can serve as a useful diagnostic marker in peripheral blood samples of children with autism spectrum disorder (ASD) [53]. In addition, the hypermethylation of a CpG site (cg20793532) in the *PPP2R2C* promoter can potentially be used as a blood biomarker for identifying adult patients with high-functioning ASD [54].

The significant alterations observed in Latvian patients with DR in the PDR/LPC group, including prolonged diabetes duration and changes in factors such as hypertension, diabetic nephropathy, cardiovascular disease, and elevated triglyceride levels, may also suggest a potential association with DNA methylation patterns [55,56]. In diabetic conditions, disruptions to metabolic homeostasis led to alterations in gene expression, affecting genes associated with oxidative stress, apoptosis, and inflammation [35,57,58]. Studies show that the duration of diabetes may correlate with changes in DNA methylation patterns [59]. They also aimed to identify specific genes and genomic regions that are affected by methylation changes associated with long-term diabetes. The Finnish Diabetic Nephropathy Study, with around 3000 diabetic patients, indicates an association between polymorphisms in the *SUV39H2* gene (which encodes histone methyltransferase) and diabetic microvascular complications, including retinopathy [60]. The development of diabetic complications may be associated with metabolic memory of previous glycemic exposure caused by epigenetic changes in target cells [61]. The higher utilization of ACEI/ARB and statin therapy observed in the Latvian PDR/LPC group may also be linked to DNA methylation levels. Qin et al. observed changes in longitudinal lipid and DNA methylation levels in the blood, after the initiation of statin treatment, with changes in DNA methylation responding primarily to changes in lipid levels rather than vice versa. Moreover, statin therapy has also been associated with changes in DNA methylation levels at certain CpGs (e.g., cg27243685 in the *ABCG1* gene) [62].

The positive correlation identified between albuminuria and methylation levels of *PSMA6* and *PSMB5* in our current study indicates a potential pathogenetic relationship between leakage of albumin following vascular damage and methylation status of these proteasome genes. According to previously published data, increased levels of global DNA methylation were also observed in diabetic patients with albuminuria compared to patients with normal albumin levels [59].

Therefore, our study’s discovery of a positive association between DNA methylation level of certain genes and DR development indicates that an elevated DNA methylation status might pose a potential risk for DR. This observation aligns with previously reported data in other populations [63,64].

Univariate regression analysis in a DR cohort of Latvian patients revealed potential associations between severe DR and *PSMA6*, *PSMB5*, and *HIF1A* gene methylation. However, adjusting for covariates in multivariate regression models (Table 4) resulted in the loss of some associations. This indicates that covariates such as age, sex, smoking, and arterial hypertension may have contributed to differences in methylation more than DR status. In addition, the diminished associations in the multivariate analysis could result from the relatively small cohort, a limitation of this study. To address this issue and enhance associations related to DR progression, future studies may consider increasing the cohort size. Several other limitations of the present investigation should be emphasized. Methylation of CpG islands often occurs in parallel with histone modifications [65]. In the present study, we focused only on methylation status and did not evaluate the histone modification profile in the regions of the genes studied; thus, histone modifications may still play a role in changes in the expression of these genes in the blood cells of diabetic patients. Moreover, the present results do not exclude epigenetic mechanisms in the upstream signaling pathways that regulate the expression of these genes or in their other regions.

The limiting condition when using the Zymo OneStep qMethyl technique to determine the methylation sites of specific genes is the number of CpG sites in the amplicon. For a region that has many CpG sites, the current method cannot provide the exact quantitation of methylation percentage without creating a greater number of primers specific to each possible methylation pattern [66]. This can become potentially costly for sequences with a large number of CpG sites. However, if one only wants to determine whether a region is highly or lowly methylated, and to compare relative levels of methylation between experimental cohorts, conventional primers can hybridize to the sequence, and this method provides a simple and relatively inexpensive way to examine this question with reasonable confidence.

We believe that this study has provided a comprehensive understanding of promoter methylation of genes related to epigenetic regulation in the process of DR. By updating our knowledge of the mechanism of methylation regulation, we can integrate these data with genetics, protein expression, and function data.

## 5. Conclusions

The exploration of epigenetics in type 1 diabetes and DR is crucial for uncovering biomarkers and therapeutic targets. Our study revealed a positive correlation between methylation patterns of the promoter regions of the proteasomal genes *PSMA6* and *PSMB5* and the hypoxia signaling axis-related gene *HIF1A* in different DR groups, linking these patterns to clinical variables. These findings suggest a potential influence of disease progression and associated factors on the methylation status of investigated genes. While our results provide insights into the interplay of epigenetic processes in these conditions, further research is warranted to enhance our understanding of the underlying mechanisms and implications for inflammatory conditions in DR.

## Figures and Tables

**Figure 1 biomedicines-12-01354-f001:**
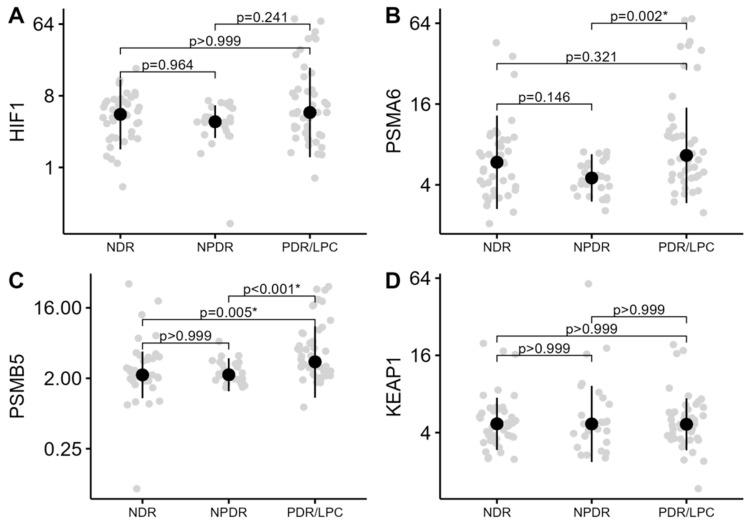
Methylation levels (%) of *HIF1A*, *PSMA6*, *PSMB5*, and *KEAP1* genes across the groups of DR severity stages. The median methylation levels (%) along with the interquartile range are highlighted. (**A**)—methylation of *HIF1A* gene; (**B**)—methylation of *PSMA6* gene; (**C**)—methylation of *PSMB5* gene; and (**D**)—methylation of *KEAP1* gene. The *y*-axis of the graphs is presented on a log scale. NDR—no diabetic retinopathy; NPDR—non-proliferative retinopathy; and PDR/LPC—proliferative retinopathy/status after panretinal-laser photocoagulation. *—statistically significant difference.

**Table 1 biomedicines-12-01354-t001:** Characteristics of patients stratified according to diabetic retinopathy status.

	NDR (*n* = 41)	NPDR (*n* = 27)	PDR/LPC (*n* = 46)	*p*-Value
Sex (female), *n* (%)	21 (51.22%)	12 (44.44%)	28 (61.87%)	0.372
Age, years	40 (30–47) ^a,b^	31 (26–43) ^b^	45 (35–51) ^a^	**0.002**
Smokers, *n* (%)	9 (21.95%) ^a^	14 (51.85%)	8 (17.39%) ^a^	**0.004**
Duration of diabetes, years	19 (13–25) ^a^	17 (13–21) ^a^	28 (22–35)	**<0.001**
Body mass index, kg/m^2^	26.60 (23.90–29.70)	22.80 (20.70–24.80) ^a^	24.40 (22.02–27.40) ^a^	**<0.001**
Waist/hip ratio	0.87 (0.77–0.94)	0.83 (0.80–0.88)	0.86 (0.78–0.95)	0.468
Arterial hypertension, *n* (%)	21 (51.22%) ^a^	12 (44.44%) ^a^	39 (84.78%)	**<0.001**
Diabetic nephropathy, *n* (%)	0 (0.00%)	6 (23.08%) ^a^	16 (35.56%) ^a^	**<0.001**
Polyneuropathy, *n* (%)	25 (60.98%)	18 (66.67%)	35 (76.09%)	0.310
Cardiovascular diseases, *n* (%)	1 (2.44%) ^a^	0 (0.00%) ^a^	10 (21.74%)	**<0.001**
HbA1c, %	8.30 (7.95–9.60) ^a^	10.15 (8.72–11.17)	8.60 (8.02–9.70) ^a^	**0.009**
Total cholesterol, mmol/L	5.32 (4.29–5.91)	4.78 (3.95–5.54)	5.20 (4.16–5.79)	0.378
High-density lipoprotein cholesterol, mmol/L	1.62 (1.26–1.91)	1.58 (1.24–1.76)	1.53 (1.21–1.90)	0.850
Low-density lipoprotein cholesterol, mmol/L	2.98 (2.33–3.42)	2.63 (1.84–3.26)	3.01 (2.07–3.34)	0.426
Triglycerides, mmol/L	0.96 (0.74–1.47) ^a^	1.09 (0.79–1.81) ^a,b^	1.32 (0.93–1.78) ^b^	**0.047**
eGFR, mL/min/1.73 m^2^	114.57 (108.26–124.13) ^a^	118.53 (108.44–128.74) ^a^	85.11 (64.17–105.30)	**<0.001**
Albuminuria, mg/mmol	0.39 (0.21–1.19)	2.58 (0.38–23.90) ^a^	3.47 (0.48–17.57) ^a^	**<0.001**
ACEI/ARB usage, *n* (%)	10 (24.39%) ^a^	6 (22.22%) ^a^	27 (58,70%)	**<0.001**
Statin usage, *n* (%)	4 (9.76%) ^a^	2 (7.41%) ^a^	15 (32.61%)	**0.006**

Continuous data are presented as medians with (Q1–Q3) with the corresponding Kruskal–Wallis test *p*-values. Categorical data are presented as frequencies (percentages) with the corresponding Chi-squared test *p*-value for the equality of proportions. ^a,b^—indicates groups that, based on pairwise post hoc comparisons, did not exhibit significant differences (according to Dunn’s or pairwise Chi-squared test). Entries with *p* < 0.05 are highlighted in bold. NDR—no diabetic retinopathy; NPDR—non-proliferative retinopathy; PDR/LPC—proliferative retinopathy/status after panretinal-laser photocoagulation; eGFR—estimated glomerular filtration rate (CKD-EPI); and ACEI/ARB—angiotensin-converting enzyme inhibitors/angiotensin II receptor blockers.

**Table 2 biomedicines-12-01354-t002:** DNA methylation levels of promoter parts of genes across the groups of different retinopathy severity stages.

	NDR	NPDR	PDR/LPC	*p*-Value
*HIF1A*, %	4.7 (2.4–6.7)	3.8 (3.1–4.9)	4.9 (2.6–9.6)	0.216
*KEAP1*, %	4.7 (3.7–5.9)	4.7 (3.6–7.1)	4.6 (3.6–5.8)	0.973
*PSMA6*, %	5.9 (3.9–8.7) ^a,b^	4.5 (3.8–5.7) ^a^	6.6 (4.7–10.7) ^b^	**0.003**
*PSMB5*, %	2.2 (1.9–3.7) ^a^	2.2 (1.9–3.0) ^a^	3.2 (2.5–7.1)	**<0.001**

The Kruskal–Wallis test was used to compare the differences in biomarkers between diabetic retinopathy groups. ^a,b^—indicates groups that, based on pairwise post hoc comparisons using Dunn’s test, did not exhibit significant differences. After adjusting for age, sex, and BMI (ANCOVA on ranks), the results remained consistent with the Kruskal–Wallis test. Entries with *p* < 0.05 are highlighted in bold. NDR—no diabetic retinopathy; NPDR—non-proliferative retinopathy; and PDR/LPC—proliferative retinopathy/status after panretinal-laser photocoagulation.

**Table 3 biomedicines-12-01354-t003:** Correlations between DNA methylation levels (%) and clinical parameters.

	NDR			NPDR			PDR/LPC		
Variable	*HIF1A*	*PSMA6*	*PSMB5*	*HIF1A*	*PSMA6*	*PSMB5*	*HIF1A*	*PSMA6*	*PSMB5*
Duration of diabetes, years	**0.52** **(<0.001)**	**0.43** **(0.005)**	−0.04 (0.804)	−0.14 (0.493)	−0.04 (0.859)	0.05 (0.797)	0.14 (0.343)	−0.08 (0.588)	−0.08 (0.616)
HbA1c, %	−0.12 (0.465)	−0.09 (0.596)	**0.44** **(0.005)**	−0.02 (0.91)	−0.08 (0.705)	−0.14 (0.483)	−0.12 (0.435)	−0.03 (0.84)	0.00 (0.993)
Albuminuria, mg/mmol	0.06 (0.728)	−0.15 (0.378)	0.24 (0.145)	0.07 (0.752)	−0.09 (0.701)	0.13 (0.554)	0.19 (0.278)	**0.45** **(0.008)**	**0.45** **(0.008)**
*KEAP1*	0.26 (0.102)	0.23 (0.145)	**0.46** **(0.003)**	0.01 (0.977)	0.26 (0.187)	−0.07 (0.741)	0.08 (0.620)	0.17 (0.272)	0.12 (0.412)
*HIF1A*		**0.55** **(<0.001)**	0.12 (0.463)		**0.39** **(0.045)**	0.15 (0.44)		**0.45** **(0.002)**	0.17 (0.26)
*PSMA6*			**0.35** **(0.024)**			−0.01 (0.977)			**0.40** **(0.006)**

Data are presented as Spearman correlation coefficient R (*p*-value). Entries with *p* < 0.05 are highlighted in bold. NDR/NPDR—no diabetic retinopathy/non-proliferative retinopathy; PDR/LPC—proliferative retinopathy/status after panretinal-laser photocoagulation.

**Table 4 biomedicines-12-01354-t004:** Results of logistic regression assessing the association between DNA methylation levels and severity of diabetic retinopathy.

Variables	Univariate Regression Results		Multivariate Regression Results ^1^		Multivariate Regression Results ^2^	
	OR (95% CI)	*p*-Value	OR (95% CI)	*p*-Value	OR (95% CI)	*p*-Value
*PSMA6*	**1.96** **(1.15; 3.33)**	**0.013**	1.01 (0.50; 2.03)	0.983	1.00 (0.57; 1,78)	0.990
*PSMB5*	**1.90** **(1.14; 3.16)**	**0.013**	1.75 (0.98; 3.12)	0.057	**1.57** **(1.04; 2.36)**	**0.030**
*KEAP1*	0.84 (0.53; 1.34)	0.472	0.77 (0.42; 1.41)	0.395	0.74 (0.45; 1.22)	0.238
*HIF1A*	**3.19** **(1.26; 8.06)**	**0.014**	2.00 (0.81; 4.93)	0.134	2.13 (0.95; 4.77)	0.065

Results of logistic regression analysis with the presence of severe diabetic retinopathy (no diabetic retinopathy and non-proliferative retinopathy vs. proliferative retinopathy/status post laser photocoagulation) as the response variable. Data are presented as odds ratios with 95% CI and *p*-values. Separate multiple regression models were fitted for each gene methylation level with the following adjustment variables: ^1^—models adjusted for age and sex; ^2^—models adjusted for age, sex, smoking, and arterial hypertension. All continuous predictors were standardized before analysis. Entries with *p* < 0.05 are highlighted in bold.

## Data Availability

The data underlying this article are available in the article and its online Appendix A.

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
