# Peer review of "DNA Methylation Profiles of PSMA6, PSMB5, KEAP1, and HIF1A Genes in Patients with Type 1 Diabetes and Diabetic Retinopathy"

_biomedicines, 2024, doi:10.3390/biomedicines12061354_

Round 1

Reviewer 1 Report

Comments and Suggestions for Authors

Review of the manuscript “DNA methylation profile of PSMA6, PSMB5, KEAP1 and HIF1A gene in patients with type 1 diabetes and diabetic retinopathy” by Zane Svikleand coauthors.

Diabetic retinopathy is one of the most prevalent complications of diabetes, affecting considerably national and global health. Despite extensive research, the precise pathogenetic mechanisms underlying diabetic retinopathy remain incompletely understood highlighting the urgent need for more effective treatments to manage its progression during the initial phases of the disease. The authors investigated differences in the DNA methylation status in several gene promoter regions in patients with type 1 diabetes and various diabetic retinopathy stages. This is an important biomedical area and the results presented in the manuscript will be interesting for the readership of the journal.

The following corrections should be made.

Abstract

Please correct the following sentence for clarity “Study included subjects with no DR (NDR, n=41), with non-proliferative DR (NPDR, n=27), with proliferative DR and post laser-photocoagulation (PDR/LPC, n=46)” as follows: ”Study subjects included individuals with no DR (NDR, n=41), those with non-proliferative DR (NPDR, n=27), and individuals with proliferative DR who underwent post-laser photocoagulation (PDR/LPC, n=46).”

Introduction:

-”In a study conducted in the US, for 2021 they estimated 9.6 million people with DR, with a prevalence rate of 26.43% among people with diabetes [1].” This is a clumsy sentence. Please, correct it as follows:” In a US study conducted in 2021, an estimated 9.6 million people were found to have DR, with a prevalence rate of 26.43% among individuals with diabetes [1].

- After the sentence “Epigenetic alterations influence how genes are expressed without making changes to the underlying DNA sequence, adapting dynamically in response to environmental, developmental, and nutritional signals [2]” please add a reference on a recent relevant review:” alpha-Synuclein and mechanisms of epigenetic regulation. Brain Sciences, 2023, 13, 150. https://doi.org/10.3390/.

-After the sentence:” The ubiquitin-proteasome system plays a crucial role in regulating cell homeostasis through protein degradation” please add a correspondent reference.

Results

“the median albuminuria level was 1.10 mg/mmol (0.28 - 8.33)”.

Please, check whether the high albuminuria level is indeed 8.33; its is unusually high.    

Discussion

“…methylation levels of genes PSMA6 and PSMB5 were found to be significantly predominant in patients with PDR/LPC”. The sense is unclear. Do you mean “significantly elevated”?

“increased proteasome gene promoter methylation” Do you mean “proteasome genes”?

Overall, this is an interesting manuscript containing new important results.

Author Response

Rebuttal letter for the Reviewer 1

We thank the reviewer for his valuable corrections and suggestions. All points have been addressed. Below we provide point-by-point answers to the issues raised (highlighted in green, see the attached file). In addition, we checked and corrected the manuscript for grammar and style mistakes.

Comments and Suggestions for Authors

Review of the manuscript “DNA methylation profile of PSMA6, PSMB5, KEAP1 and HIF1A gene in patients with type 1 diabetes and diabetic retinopathy” by Zane Svikle and coauthors.

Diabetic retinopathy is one of the most prevalent complications of diabetes, affecting considerably national and global health. Despite extensive research, the precise pathogenetic mechanisms underlying diabetic retinopathy remain incompletely understood highlighting the urgent need for more effective treatments to manage its progression during the initial phases of the disease. The authors investigated differences in the DNA methylation status in several gene promoter regions in patients with type 1 diabetes and various diabetic retinopathy stages. This is an important biomedical area and the results presented in the manuscript will be interesting for the readership of the journal.

The following corrections should be made.

Abstract

Please correct the following sentence for clarity “Study included subjects with no DR (NDR, n=41), with non-proliferative DR (NPDR, n=27), with proliferative DR and post laser-photocoagulation (PDR/LPC, n=46)” as follows: ”Study subjects included individuals with no DR (NDR, n=41), those with non-proliferative DR (NPDR, n=27), and individuals with proliferative DR who underwent post-laser photocoagulation (PDR/LPC, n=46).” ]. - the sentence has been corrected correspondingly

Introduction:

-”In a study conducted in the US, for 2021 they estimated 9.6 million people with DR, with a prevalence rate of 26.43% among people with diabetes [1].” This is a clumsy sentence. Please, correct it as follows:” In a US study conducted in 2021, an estimated 9.6 million people were found to have DR, with a prevalence rate of 26.43% among individuals with diabetes [1]. - the sentence has been corrected correspondingly

- After the sentence “Epigenetic alterations influence how genes are expressed without making changes to the underlying DNA sequence, adapting dynamically in response to environmental, developmental, and nutritional signals [2]” please add a reference on a recent relevant review:” alpha-Synuclein and mechanisms of epigenetic regulation. Brain Sciences, 2023, 13, 150. https://doi.org/10.3390/.

Thank you for mentioning this nice paper. Citation: Surguchov, A (2023) α-Synucleinand Mechanisms of EpigeneticRegulation. Brain Sci. 13,150. https://doi.org/10.3390/brainsci13010150- has been added

-After the sentence:” The ubiquitin-proteasome system plays a crucial role in regulating cell homeostasis through protein degradation” please add a correspondent reference.

- corresponding references have been added at the end of this sentence. These are:

Chen, Y.,Hong, T., Wang, S., Mo, J., Tian, T., and Zhou, X. (2017) Epigenetic modification of nucleic acids: from basic studies to medical applications. Chem. Soc. Rev., 46,2844–2872.

Surguchov, A. (2023) α-Synuclein and Mechanisms of Epigenetic Regulation. Brain Sci., 13 (1):150.

Results

“the median albuminuria level was 1.10 mg/mmol (0.28 - 8.33)”.

Please, check whether the high albuminuria level is indeed 8.33; its is unusually high.   

These median numbers correspond to normo- and microalbuminuria for albumin/creatinine ratio in urine portion (normoalbuminuria - <3 mg/mmol; microalbuminuria 3-30 mg/mmol, macroalbuminuria>30 mg/mmol, from ADA, Diabetes Care, 2019 Jan;42 (Supplement1): S103-S138). As there are patients with severe retinopathy in our cohort, such values are awaited. Moreover, as seen in Table 1, there are patients with macroalbuminuria and eGFR below 60 ml/min/1.732 in NPDR and PDR/LPC groups (n=6(23.08%) and n=16 (35.56%) respectively).

Discussion

“…methylation levels of genes PSMA6 and PSMB5 were found to be significantly predominant in patients with PDR/LPC”. The sense is unclear. Do you mean “significantly elevated”?

this statement has been clarified -“…methylation levels of genes PSMA6 and PSMB5 were found to be significantly higher in patients with PDR/LPC”.

“increased proteasome gene promoter methylation” Do you mean “proteasome genes”? –

the sentence has been clarified - “higher proteasome gene promoter methylation’’. The grammar check confirmed that it is correct to write “proteasome gene promoter methylation’’

Overall, this is an interesting manuscript containing new important results.

Reviewer 2 Report

Comments and Suggestions for Authors

Dear Authors,

This is very interesting manuscript, and I have only very little suggestions

1.       Lines 40-42- I agree that is lack of indicators, but in Your study You want to reveal rather susceptibility, probablity. And  off course we don’t know if such methylation profile is characteristic for DR, or for other microangiopathic  diabetes complication too (what in correlations You indicated). Please change these sentenses accordingly.

2.       Line 90- Notable references include- - What for such sentence?

3.       Line 106- Our study aimed to study…- It is not gramaticaly proper

4.       Lines 126-132- Maybe better understandible will be at start said that Group of Proliferative DR include patients after laserotherapy.

5.       Statistical analysis- there is lack of sentence- what level was recognized as significant

6.       Statistical software- Lack of producer, or link to version of statistical program

7.        Discussion- line 346-347- Correlation HbA1c with NDR- but why not with  groups with DR- this is not logical, please try to explain

Author Response

Rebuttal letter for the Reviewer 2

We thank the reviewer for his valuable corrections and suggestions. All points have been addressed. Below we provide point-by-point answers to the issues raised (highlighted in green, please see the attached file). In addition, we checked and corrected the manuscript for grammar and style mistakes.

Comments and Suggestions for Authors

Dear Authors,

This is very interesting manuscript, and I have only very little suggestions

  1. Lines 40-42- I agree that is lack of indicators, but in Your study You want to reveal rather susceptibility, probablity. And off course we don’t know if such methylation profile is characteristic for DR, or for other microangiopathic  diabetes complication too (what in correlations You indicated). Please change these sentenses accordingly.

this statement has been clarified –

  1. The pathogenetic mechanisms responsible for DR are not yet fully understood. At present, there is a deficiency in identifying susceptibility and probability of early indicators for DR and other microvascular complications of diabetes, and there is also a need for effective treatments to manage the progression of the disease during its initial stages.
  2. Line 90- Notable references include- - What for such sentence?

this statement has been clarified –

In experimental studies, inhibitors of proteasome activity have shown a beneficial impact on the progression of diabetic complications [23–26].

  1. Line 106- Our study aimed to study…- It is not gramaticaly proper

this statement has been clarified –

Our study aimed to investigate the differences in the methylation status of the promoter parts of PSMA6, PSMB5, HIF1A, and KEAP1 genes between patients with different DR 107 stages and type 1 diabetes in Latvia.

  1. Lines 126-132- Maybe better understandible will be at start said that Group of Proliferative DR include patients after laserotherapy.

The paragraph was corrected: DR grading was based on the fundus oculi examination carried out by an ophthalmologist. Patients were stratified according to the DR status as follows: no retinopathy (NDR), non-proliferative retinopathy (NPDR), proliferative retinopathy (PDR), patients after panretinal-laser photocoagulation (LPC). Further, to increase the power of the study, subjects were stratified into three groups: patients with no signs of DR were marked as no DR (NDR), patients with NPDR; patients with PDR and those after LPC were included in the PDR/LPC group.

  1. Statistical analysis- there is lack of sentence- what level was recognized as significant

We added a sentence to the methods section, statistical analysis: “A p-value of less than 0.05 was considered statistically significant.”

  1. Statistical software- Lack of producer, or link to version of statistical program

We checked the version of the programme used and corrected the sentence: “All statistical data analysis was performed using Statistical Software R version 4.3.0. (http://www.r-project.org), last accessed on April 23rd 2024.

  1. Discussion- line 346-347- Correlation HbA1c with NDR- but why not with groups with DR- this is not logical, please try to explain.
  2. Thank you very much for this interesting comment. We try to explain here: PSMB5 methylation is just one of many potential epigenetic modifications that can occur, and its effect may be influenced by other factors or modifications. In the context of diabetes, it is possible that PSMB5 methylation is more sensitive to early changes in blood glucose levels, such as those that occur in the NDR group, where individuals may be experiencing impaired fasting glycaemia or prediabetes. In the DR groups, where individuals have already developed retinopathy, other genetic, environmental, or lifestyle factors may be playing a more significant role in disease progression and the development of complications. These factors could be overriding or interacting with the effects of PSMB5 methylation, making the correlation less apparent. Additionally, the sample size and demographic characteristics of the cohort may also play a role. A larger or more diverse sample could reveal subtler correlations that may not be evident in a smaller or more homogeneous group.

We added a sentence to the corresponding discussion part: “Interestingly, no correlation between HbA1c and PSMB5 gene methylation was observed in NPDR and PDR/LPC groups, which could be explained by confounding factors: higher levels of HbA1c, higher prevalence of other complications of diabetes, and more frequent usage of antihypertensive and hypolipidaemic treatment in these groups [41-46].”
